# Effect of Guanidinoacetic Acid and Zilpaterol Hydrochloride Feed Additions on Lambs’ Productive Performance, Carcass Characteristics, and Blood Chemistry

**DOI:** 10.3390/ani15121692

**Published:** 2025-06-07

**Authors:** Daniel López-Aguirre, Javier Hernández-Meléndez, José F. Vázquez-Armijo, Luz Y. Peña-Avelino, Jorge Alva-Pérez

**Affiliations:** 1Facultad de Ingeniería y Ciencias, Universidad Autónoma de Tamaulipas, Ciudad Victoria 87149, Mexico; dlaguirre@docentes.uat.edu.mx (D.L.-A.); javhernan@docentes.uat.edu.mx (J.H.-M.); 2Centro Universitario UAEM Temascaltepec, Universidad Autónoma del Estado de México, Temascaltepec 513000, Mexico; jfvazqueza@uaemex.mx; 3Facultad de Medicina Veterinaria y Zootecnia “Dr. Norberto Treviño Zapata”, Universidad Autónoma de Tamaulipas, Ciudad Victoria 87274, Mexico; lypena@docentes.uat.edu.mx

**Keywords:** feed additive, meat quality, energy metabolism, glucose, creatine

## Abstract

The present study evaluates the effects of two dietary supplements (guanidinoacetic acid [GAA] and zilpaterol hydrochloride [ZLH]) on the productive performance of fattening lambs over a 60-day period. The inclusion of these additives did not result in significant differences in the parameters studied. However, serum glucose and creatinine levels in the GAA group were higher than those in the control group. These findings suggest that further research is needed to determine the optimal dosage and duration of GAA supplementation to enhance lamb growth and meat production.

## 1. Introduction

Feed additives for animals have experienced high demand in emerging economies [1]. The use of additives in livestock has shown beneficial effects on physiological, productive, and health parameters [2]. The use of growth promoters in animal production yields varying results, with zilpaterol hydrochloride (ZLH) being one of the most extensively studied feed additives, whereas guanidinoacetic acid (GAA) has been explored to a minor extent.

GAA, also known as glycinamide (CAS No. 352-97-6), is synthesized in the liver, kidneys, and pancreas from L-arginine and glycine. It is then methylated by S-adenosylmethionine to produce creatine, which plays a crucial role in muscle energy metabolism and protein synthesis—both essential for rapid growth in animals [3,4]. Phosphocreatine, derived from creatine, provides energy for quick muscle contractions. Supplementing with exogenous GAA can boost creatine synthesis and enhance energy availability in muscle cells [5]. GAA also participates in energy metabolism regulation and may influence hormonal modulation and antioxidant activity, although these effects need further study [6]. GAA used as a dietary additive for growing and finishing pigs has been shown to improve carcass quality by reducing drip loss and the yellowing of meat [7]. It also enhances bone growth and development in poultry, which is significant since broilers cannot synthesize the amino acid arginine [3]. For Angus bulls, supplementing with 0.6 to 0.9 g kg^−1^ DM of GAA has improved growth performance, nutrient digestion, and ruminal fermentation [8]. In crossbred steers (Bos taurus x Bos indicus), adding GAA at 1 g kg^−1^ of total mixed diet reduced dry matter intake and improved the feed conversion efficiency, though it did not affect body weight or daily weight gain [9]. Numerous studies show that GAA benefits fattening lambs by improving productive performance, antioxidant capacity, and metabolizable energy availability [10,11,12,13,14,15]. Notably, Jin et al. found that at 1000 mg kg^−1^, GAA increased muscle fiber and reduced meat drip loss in Hu lambs, but also raised muscle shear force, potentially impacting meat quality [15].

ZLH is a β-adrenergic agonist originally developed as a feed additive for beef cattle. These compounds are organic molecules that bind to β-adrenergic receptors located on cell membranes, leading to a decrease in lipogenesis (fat synthesis and storage) and an increase in lipolysis (fat mobilization and hydrolysis) [16,17]. In fattening lambs, ZLH supplementation during the last 28 to 30 days of the finishing period has been reported to increase the dressing percentage, the hot carcass weight, and the muscle area of the longissimus dorsi and longissimus thoracis et lumborum [18,19,20]. In addition, feed efficiency, growth rate, and dietary net energy were increased in finishing lambs supplemented with ZLH [18,21,22]. These productive characteristics are important economic parameters that ZLH could improve.

A recent study by [22] showed that the strategic use of calcium propionate and ZLH can optimize lamb meat productivity. Supplementation with these two additives resulted in an increase in the final body weight, the average daily gain, and the dressing percentage of Dorper lambs. However, the improvements did not follow a linear pattern, with the optimal effects observed at moderate levels of supplementation. It is hypothesized that both GAA and ZLH, when used as dietary supplements in fattening lambs, can enhance productivity. While ZLH has been extensively studied in ruminants, particularly cattle, research on GAA in ruminants is more recent and limited. This study aimed to evaluate the individual effects of dietary supplementation with GAA and ZLH on growth performance, carcass yield, meat color, and blood chemistry profiles, compared to a control group.

## 2. Materials and Methods

The experiment was conducted at the Animal Production unit of the “Ingeniero Herminio García González” Animal Science Farm in the Faculty of Engineering and Sciences (FIC) at the Autonomous University of Tamaulipas (UAT), located in the municipality of Güémez, Tamaulipas, Mexico (23°56′ N, 99°06′ W). The climate is classified as dry tropical, characterized as semi-arid and sub-humid. The average annual temperature is 23 °C with a total annual rainfall of approximately 800 mm. Laboratory analyses, as well as sample conservation and processing, were carried out at the Animal Nutrition Laboratory in the FIC-UAT, located in the Adolfo López Mateos University Center in Ciudad Victoria, Tamaulipas, Mexico.

### 2.1. Experimental Design, Animals, Diet, and Feeding Management

All animal procedures were approved by the Institutional Animal Welfare Committee (authorization number: CBBA-17-21). Twenty-four non-castrated male crossbred Pelibuey × Dorper lambs (2.5 months old; average live weight: 16.3 ± 2.7 kg) were housed in individual pens (1.5 m × 0.75 m) on a concrete floor. During the initial weighing, the animals were fasted for 24 h. The animals underwent a 14-day adaptation period before the experimental study began. The experiment lasted for 60 days. Each pen was equipped with a feeder, water container, and shade. In addition, 1 mL of vitamins ADE (Vigantol; Elanco, Zapopan, Mexico) and 0.5 mL of ivermectin, a drug for the treatment of internal and external parasites (Ivermectin; Sanfer Laboratory, Mexico City, Mexico), were administered to each animal on the first day of the adaptation period. A total mixed ration (TMR) was formulated to meet the nutritional requirements of the lambs with an expected daily weight gain of 250 to 300 g/day (Table 1), following the NRC recommendations [23]. Before the onset of the experiment, each lamb was weighed and randomly assigned to one of three experimental treatments (*n* = 8) with each animal considered an experimental unit. The treatments were as follows: (1) Control: TMR diet without the inclusion of any additive, (2) GAA: TMR diet with the inclusion of GAA (GuanAMINO®, Evonik, Essen, Germany, provided in CaCO_3_ at 0.06% of the DM of the diet; depending on intake, this corresponds to a dose of 0.6 g GAA/animal) [8], and (3) ZLH: TMR diet with ZLH (Tairui Pharmaceutical CO., LTD, Ningxia, China, administered at 6 mg/kg of DM during the last 30 days of the experimental phase [18]). TMR was offered at 08:00 and 16:00 h, in proportions of 60% and 40% of the total daily intake, respectively. The amounts of TMR provided were adjusted to ensure ad libitum consumption (allowing for approximately 10% refusal of DM intake). All animals had ad libitum access to fresh water.

### 2.2. Body Weight and Animal Performance

The body weight (BW) of each experimental unit was recorded before morning feeding on days 1, 15, 30, 45 and 60 of the experiment. The diet provided to the animals was adjusted based on 4% of BW. The amount of feed offered and rejected was recorded daily. If necessary, the feed offered was adjusted based on the intake from the previous day. From the collected data, the average daily gain, total weight gain, feed conversion ratio (kg fed/kg gain), and feed efficiency (g of BW/kg DM) were calculated.

### 2.3. Chemical Analyses

All dehydrated feed samples were ground to a 2 mm particle size (using a Wiley mill [Thomas Wiley, Laboratory Mill model 4, Thomas Scientific TM, Swedesboro, NJ, USA]). The samples were analyzed to determine dry matter (DM), in a forced air oven (Luzeren, Proveedor de Laboratorios, SA de CV, Tlajomulco de Zuñiga, Mexico) at 105 °C for 24 h; Method 930.15), organic matter (OM, by weight loss after calcination in a muffle [F48015, Barnstead/thermolyne, Dubuque, IA, USA]) at 550 °C for 6 h; Method 942.05), and nitrogen (N, Kjeldahl procedure; Method 954.01), following the methods described by the AOAC [24]. Crude protein (CP) content was calculated by multiplying N by 6.25. The content of neutral detergent fiber (NDF, with thermostable α-amylase and sodium sulfite) and acid detergent fiber (ADF) [25] were determined using ANKOM F-57 filter bags on an ANKOM^200^ fiber analyzer (ANKOM Technology, Macedon, NY, USA).

### 2.4. Carcass Characteristics and Non-Meat Components

Four animals per treatment were slaughtered immediately after the 60-day feeding period. Feed and water were withdrawn 12 h before the procedure. The carcasses were bled, skinned, and eviscerated to determine the hot carcass weight (HCW). After 24 h of cooling at 4 °C, the cold carcass weight (CCW), carcass length, thorax depth, leg length, and leg perimeter were recorded. Additionally, the pH was measured 24 h postmortem using a portable digital pH meter (HI 99163, Hanna Instruments, Vöhringen, Germany) in the longissimus thoracis et lumborum (LTL) muscle. The weights of the non-meat components (blood, liver, and skin) were expressed as a percentage of the final carcass weight. The cooling loss percentage was calculated using the following formula: CL% = ([HCW-CCW]/HCW) × 100. The dressing percentage was calculated as: D% = ([HCW/BW at day 60] × 100). The following cuts were obtained from the cold carcass: neck, leg, ribs/skirt, loin, and forequarter/shoulder, according to the specifications in [26,27]. The yield of each cut was expressed as a percentage of the CCW.

### 2.5. Meat Color Evaluation

Color was measured on the surface of the exposed LTL muscle at the 12th/13th rib section. Measurements were performed using a Minolta CR-400 colorimeter (Konica Minolta Sensing, Inc., Osaka, Japan) with D65 illuminant. Color parameters L* (lightness), a* (redness), and b* (yellowness) were evaluated according to [28]. Hue angle (Hue) was calculated using the formula: Hue = tan^−1^ (b*/a*), and Chroma (color saturation index) using the formula: Chroma = (a*^2^ + b*^2^)^1/2^ [21]. Color measurements were taken at different locations on the surface (perpendicular to the muscle fibers) of the cold samples (4 °C).

### 2.6. Sample Collection and Analysis of Serum Biochemical Parameters

Blood samples were collected from all animals in each treatment group. Blood samples were obtained before morning feeding from the jugular vein. Sampling occurred on days 1, 15, 30, 45, and 60 of the experiment. The samples were collected in tubes without anticoagulant (BD Vacutainer, Franklin Lakes, NJ, USA) and processed by centrifugation at 1500× *g* for 20 min to obtain serum. Serum samples were stored at −20 °C prior to analysis. Serum samples were used for the determination of glucose, cholesterol, triglycerides, bilirubin, creatinine and blood urea nitrogen (BUN) by semi-automated chemical analyzer (Model BA-88A Mindray, Shenzhen, China). Measures were performed using Spinreact kits (Spinreact, Girona, Spain) following the manufacturer’s instructions.

### 2.7. Statistical Analysis

The Shapiro-Wilk test was applied to assess normality, while homoscedasticity was evaluated using Bartlett’s test. Animal performance, carcass parameters, serum parameters, and meat color data were analyzed using the mixed procedure of SAS (version 9.2) [29]. Means and standard errors were obtained using least squares, and multiple means comparisons were conducted with Tukey’s test. Orthogonal contrasts were performed to determine the effect of the addition of the additive inclusion (contrast 1: Control versus GAA, ZLH) and the differences between additive treatments (contrast 2: GAA versus ZLH). Significant differences were declared at *p* < 0.05. The covariance structure was selected based on the lowest Akaike and Schwarz Bayesian criteria values among the tested structures. The statistical model used for the analysis was as follows:
$$Y_{\mathrm{ij}}=\mu T_{i}+\beta (X_{\mathrm{ij}}-\bar{X})+\varepsilon_{\mathrm{ij}}$$
where Y_ij_ is the observed value, μ is the overall mean, T_i_ is the fixed effect of treatment, X_ij_ is the covariate (initial body weight, hot carcass weight, or cold carcass weight), X is the overall mean of the covariate, β is the regression coefficient of Y on X, and ε_ij_ is the residual error.

## 3. Results

### 3.1. Animal Performance and Serum Biochemical Parameters

Initial body weight (BW), final BW, total weight gain (TWG), average daily gain (ADG), dry matter intake (DMI), feed conversion ratio (FCR), and feed efficiency (FE) did not differ between groups (Table 2). Similarly, no changes were observed based on contrast 1 (control versus additives; *p* > 0.05) or contrast 2 (GAA versus ZHL; *p* > 0.05). The effects of ZLH finishing and GAA additive inclusion on blood biochemistry are presented in Table 2. There were no differences in cholesterol, triglycerides, bilirubin, and BUN between the GAA and ZLH groups. Serum glucose and creatinine levels were higher in the treatments with additives compared to the control group (contrast 1; *p* ≤ 0.05), but no differences were observed between GAA and ZLH (contrast 2; *p* > 0.05).

### 3.2. Carcass Characteristics, Meat Color, and Non-Meat Components

Carcass characteristics did not differ among the groups receiving additives (*p* > 0.05, Table 3). Furthermore, no statistical differences were observed in contrast 1 (control versus additives; *p* > 0.05) or contrast 2 (GAA versus ZLH; *p* > 0.05). Meat components and pH measured at 24 h postmortem were also not affected by the dietary treatments (*p* > 0.05), except for the forequarter and shoulder weights, which were significantly different (*p* = 0.0233). The percentage forequarter and shoulder cuts differed in contrast 1 (control versus additives; *p* < 0.05), but no differences were observed in contrast 2 (GAA versus ZLH; *p* > 0.05). Non-meat components (NMC) did not differ among the dietary groups. Similarly, no significant differences were detected in any of the contrasts (*p* > 0.05; Table 3).

In the meat color parameters, the GAA and ZLH groups did not affect the lightness, redness, yellowness, chroma, or hue angle of the lamb meat (*p* > 0.05; Table 4). Moreover, no significant differences were observed in any of the contrasts (*p* > 0.05).

## 4. Discussion

### 4.1. Body Weight and Animal Performance

The addition of GAA to the diet has been suggested to improve energy metabolism [12] and rumen fermentation, which facilitates nutrient digestion [30]. In this study, the GAA and ZLH additives did not affect the productive performance of the animals. This is similar to results obtained by Majdeddin et al. [31], who evaluated different doses of GAA (0, 0.6, and 1.2 g/kg) in poultry. Similarly, Zhu et al. [32] did not observe changes in the feed intake, average daily gain, or feed conversion ratio in the diets of growing and finishing pigs. On the contrary, Li et al. [33] observed an increase in the total weight gain when 0.2% GAA was included in the diet of Jinjiang bulls, whereas no improvement was seen at inclusion levels of 0.4%, 0.1%, or 0.05%. Notably, the best feed conversion ratio was obtained at the 0.2% inclusion level. Zhu et al. [32] and Li et al. [33] mention that the difference in the results obtained with GAA could be due to variations in the following conditions: age, sex, species, study duration, nutrient levels, and the difference in dietary amino acids, particularly methionine and cysteine. In this context, Córdova-Noboa et al. [34] observed a better feed conversion ratio and higher BW in broilers when 0.06% GAA and 5% poultry by-products were included in the diet. These authors suggest that the effect of GAA is observed when it is included in diets containing animal-origin meals. Therefore, it is possible that the differences with our results are partly due to the fact that the diets were formulated with vegetable-origin protein, and the dose used in the diet of intensive-fattening lambs was lower than the studies reporting productive performance effects with GAA.

### 4.2. Serum Biochemical Parameters

Blood biochemical parameters were within the reference intervals [35], suggesting that the addition of the additives (GAA and ZLH) did not alter the health status of finishing lambs. In this study, a higher serum glucose level was observed when GAA was included at 0.06% of DM, compared with the control group. Also, the inclusion of the ZLH additive resulted in a similar increase in serum glucose levels as observed in the GAA group. In contrast to our findings, ref. [11] did not observe an increase in glucose levels when GAA was included for Dorper × Han ram lambs. Also, ref. [12] reported similar serum glucose results to those of the latter study in Chinese Han lambs. Volatile fatty acids (VFA) account for almost 70% of the energy requirement in ruminants [36]. It is possible that the glucose increase observed in our study was due to GAA administration, which may have promoted an increase in ruminal VFA production. It has been reported that GAA supplementation promotes an increase in VFA levels in lambs [11,13].

In this study, creatinine values for all diets were within the reference range (1.0–2.7 mg/dL [35]. An increase in serum creatinine was observed in the GAA group compared to the control group. This increase in creatinine levels in the GAA group is similar to the observations of [8], who tested different doses of GAA in bulls [33]. Likewise, ref. [33] observed that with doses ranging from 0.05% to 0.4%, creatine levels increased compared to the control group in Jinjiang bulls. With respect to fattening lambs, elevated creatinine levels were observed compared to the control group [11], in agreement with the findings in this study. The increase in creatinine levels may enhance nutrient digestibility at the ruminal level, increase the concentration of VFAs, and raise the proportion of propionate, which favors energy metabolism. This increase could be explained by the fact that GAA is an immediate precursor of creatine. It is methylated at the amidino group by S-adenosyl-methionine (SAM) via the guanidinoacetic methyltransferase pathway and subsequently phosphorylated to phosphocreatine [37]. Although no improvement in productive performance was observed, the increase in serum glucose and creatinine concentrations in the GAA group may indicate a shift in energy metabolism associated with GAA supplementation. These findings support the need for further research to evaluate the effects of extended feeding periods or different inclusion levels of this additive.

### 4.3. Carcass Characteristics and Non-Meat Components

The use of ZLH in animal production has been documented in the literature. Several studies have shown that the inclusion of ZLH improves fattening performance and carcass characteristics in cattle [38] and lambs [39]. The beta-adrenergic agonist ZLH was originally developed for feedlot cattle finishing, showing consistent and favorable results on growth performance and certain carcass characteristics, but with little or no effect on meat quality [40,41,42]. However, when this product is used in small ruminants, although most results are positive, the exact dose of ZLH that maximizes the growth rate and carcass characteristics is not well defined, as studies have not reported consistent findings [17,21,22]. In the present study, the addition of GAA or ZLH feed additive during the last 30 days of the experiment had no effect on the carcass characteristics of fattening lambs. It possible that the inclusion of the ZLH diet inclusion at 6 mg/kg of DM in the diet was insufficient to produce the expected effect reported in several studies [43,44,45,46].

### 4.4. Meat Color

The inclusion of GAA at 0.06% of DM or ZLH at 6 mg/kg in the diet of fattening lambs did not affect the color of the lamb’s meat [38]; in their study with Nerole bulls and steers, 7.5 mg of ZLH was incorporated into the diet for 20 days and they found no differences in meat color, which is consistent with the present study. On the other hand, ref. [19] observed differences in the meat color parameters (L*, a*, and C) with increasing levels of ZLH (0.1, 0.2, and 0.3 mg/kg) administered for 30 days during the finishing stage of the lambs. These authors explained that the incorporation of ZLH resulted in a reduction in luminosity, and the loss of luminosity can have negative consequences for consumer acceptance of the product [47]. However, this effect was not observed in the present study, which may be due to the low levels of additives incorporated into the diet.

## 5. Conclusions

The inclusion of GAA or ZLH during the finishing stage of fattening lambs (Pelibuey × Dorper), did not improve the carcass characteristics or meat color. The results of this study suggest that extending the duration of the trial and/or using different additive dosages may be necessary to achieve the desired productive responses.

## Figures and Tables

**Table 1 animals-15-01692-t001:** Ingredients and chemical composition of experimental diet (%).

Ingredients	% DM
Soybean	14.5
Corn, ground grain	10
Sorghum, ground grain	39
Sorghum, whole grain	10
Alfalfa, hay	11
Sorghum, hay	7.5
Molasses	5
Minerals	2.5
CaCO_3_	0.5
Total	100
**Nutrients**	
Crude protein	15.2
Neutral detergent fiber	20.98
Acid detergent fiber	12.13
Ash	4.42
Ether extract	2.61
Metabolizable energy (Mcal/kg DM)	2.78

**Table 2 animals-15-01692-t002:** Growth performance, dry matter intake, feed conversion rate, and biochemical parameters in lambs fed diets with GAA or ZLH.

Items ^1^	Treatments ^2^			SEM	*p*-Value ^3^		
	CON	ZLH	GAA		T	C1	C2
Initial BW, kg	15.72	16.31	16.88	1.047	0.7363	0.5272	0.6533
Final BW, kg	29.45	30.91	30.97	1.365	0.6757	0.3818	0.9745
TWG, kg	13.71	14.70	14.11	0.711	0.6213	0.4351	0.5657
ADG g/day	240.50	257.75	247.62	12.535	0.6265	0.4362	0.5740
FI, g/day	1017.87	1057.25	1052.12	68.924	0.9084	0.6672	0.9586
FCR, kg/kg	4.24	4.09	4.30	0.2578	0.8491	0.8958	0.5823
FE, g/kg	212.63	256.54	246.53	26.190	0.4746	0.2386	0.7896
GLU mg/dL	59.78 ^b^	62.33 ^ab^	70.26 ^a^	2.471	0.0007	0.0032	0.5022
CHO, mg/dL	46.48	45.34	45.78	2.190	0.8930	0.9601	0.7119
TGL, mg/dL	18.33	17.36	15.43	1.238	0.1966	0.1140	0.5809
BIL, mg/dL	0.3012	0.266	0.243	0.020	0.1779	0.1103	0.2347
CREA, mg/dL	1.21 ^b^	1.25 ^ab^	1.34 ^a^	0.039	0.0365	0.0271	0.4040
BUN, mg/dL	19.08	19.65	20.66	0.748	0.1456	0.1623	0.5892

^1^ BW: body weight; TWG: total weight gain; ADG: average daily gain; FI: feed dry matter intake; CR: feed conversion ratio (kg feed/kg gain); FE: feed efficiency (kg of BW/kg DM); GLU: glucose; CHO: cholesterol; TGL: triglycerides; BIL: bilirubin; CREA: creatinine; BUN: blood urea nitrogen. Means with different letters indicate statistical differences. ^2^ CON: diet without additives; ZLH: TMR with zilpaterol hydrochloride at 6 mg/kg of DM during the last 30 days of the experiment; GAA: diet with guanidinoacetic acid at 0.06% of dry matter. ^3^ T: fixed effect of treatment; C1: contrast 1 “control versus additives”; C2: contrast 2 “ZHL versus GAA”. SEM: standard error of the mean.

**Table 3 animals-15-01692-t003:** Carcass characteristics and non-meat components in lambs fed diets with GAA or ZLH.

Items ^1^	Treatments ^2^			SEM	*p*-Value ^3^		
	CON	ZLH	GAA		T	C1	C2
**Carcass characteristics**							
Final BW, kg	30.63	31.62	31.82	1.480	0.8350	0.5641	0.9245
HCW, kg	15.66	16.38	16.49	0.780	0.7303	0.4444	0.9214
CCW, kg	15.13	16.00	15.85	0.720	0.6721	0.3818	0.8860
D, %	51.10	51.86	51.77	0.680	0.7226	0.4120	0.9283
CL, %	3.33	2.32	3.73	0.980	0.5938	0.8007	0.3350
Neck, % of CCW	8.41	8.85	9.89	0.669	0.3225	0.2719	0.3011
Leg, % of CCW	23.53	23.73	24.00	0.672	0.8828	0.6917	0.7772
Rib and flank, % of CCW	23.52	21.20	20.80	0.620	0.3619	0.2106	0.5132
Loin, % of CCW	23.90	25.09	23.94	0.449	0.1591	0.2940	0.1027
Forequarter and shoulder, % of CCW	19.91 ^a^	17.79 ^b^	19.18 ^ab^	0.444	0.0233	0.0281	0.0536
Carcass length, cm	62.75	64.75	65.75	1.920	0.5520	0.3146	0.7207
Thorax depth, cm	17.12	15.63	17.25	0.994	0.4683	0.5863	0.2778
Leg perimeter, cm	38.25	39.25	41.00	1.370	0.3945	0.2925	0.3897
Leg length, cm	44.25	45.25	44.25	1.870	0.9102	0.8323	0.7145
pH at 24 h	5.77	5.68	5.71	0.084	0.7456	0.4723	0.8387
**NMC**							
NMC, % BW	48.30	47.52	47.95	0.880	0.8232	0.6095	0.7394
Blood, % of BW	4.30	4.56	4.35	0.269	0.7692	0.6431	0.5909
Liver, % of BW	2.34	2.12	2.18	0.129	0.6666	0.3971	0.8140
Skin, % of BW	9.19	8.93	9.89	0.599	0.5239	0.7666	0.2842

^1^ Final BW: final body weight; HCW: hot carcass weight; CCW: cold carcass weight; D: dressing percentage; CL: cooling loss percentage; NMC: non-meat components. Means with different letters indicate statistical differences. ^2^ CON: diet without additives; ZLH: TMR with zilpaterol hydrochloride at 6 mg/kg of DM during the last 30 days of the experiment; GAA: diet with guanidinoacetic acid at 0.06% of dry matter. ^3^ T: fixed effect of treatment; C1: contrast 1 “control versus additives”; C2: contrast 2 “ GAA versus ZLH”. SEM: standard error of the mean.

**Table 4 animals-15-01692-t004:** Meat color quality of Longissimus dorsi of lambs feed diets with GAA or ZLH.

Items ^1^	Treatments ^2^			SEM	*p*-Value ^3^		
	CON	ZLH	GAA		T	C1	C2
L*	44.14	42.22	40.79	1.403	0.2865	0.1586	0.4898
a*	16.61	16.35	16.24	0.476	0.8545	0.6025	0.8694
b*	7.71	6.43	5.60	0.705	0.1588	0.0812	0.4278
C	18.37	17.58	0.66	0.605	0.4090	0.2159	0.6607
H	24.62	21.52	19.08	1.875	0.1670	0.0923	0.3809

^1^ L*: lightness; a*: redness; b*: yellowness; C: chroma; H: hue angle. ^2^ CON: diet without additives; ZLH: TMR with zilpaterol hydrochloride at 6 mg/kg of DM during the last 30 days of the experiment; GAA: diet with guanidinoacetic acid at 0.06% of dry matter. ^3^ T: fixed effect of treatment; C1: contrast 1 “control versus additives”; C2: contrast 2 “GAA versus ZLH”. SEM: standard error of the mean.

## Data Availability

The data supporting this study are openly available in 10.6084/m9.figshare.24274915, as datasets.

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
