# Peer review of "Effect of Guanidinoacetic Acid and Zilpaterol Hydrochloride Feed Additions on Lambs’ Productive Performance, Carcass Characteristics, and Blood Chemistry"

_animals, 2025, doi:10.3390/ani15121692_

Round 1
Reviewer 1 Report
Comments and Suggestions for Authors
The study is interesting because additives like zilpaterol and guanidoacetic acid have been utilized to enhance animal productivity. However, some concepts could be more precise, particularly the difference between creatinine and creatine. These are distinct compounds, yet they are often referenced as the same.

While it has been well written, the English needs some improvement.
Author Response
Response to general comments
We appreciate the reviewer comment and acknowledgment of the study’s relevance. We carefully reviewed the manuscript and clarified the concepts to ensure a clear distinction between creatine and creatinine. These revisions were implemented in the introduction and discussion sections to prevent any misunderstanding. We carefully reviewed the experimental design section and have clarified specific aspects of the methodology to ensure transparency and reproducibility. We reviewed the conclusions to ensure they are fully supported by the results. Minor adjustments were made to the wording in each section. The manuscript has been revised to improve the quality of English and ensure clarity, precision, and readability the text.
Introduction
- Comment: Are you referring to creatine and phosphocreatine? Creatinine is a byproduct of creatine breakdown, not an energy source.
Response: We reviewed and clarified the terminology in the introduction to accurately distinguish between creatine/phosphocreatine and creatinine. These corrections were made directly in the manuscript; however, we note that the changes were applied with track changes turned off
Material and methods
- Comment: Why did you not have a treatment just with zilpaterol?
Response: We clarify that the study did include a treatment group with zilpaterol hydrochloride (ZLH) alone. The study included three treatments: a control group (no additives), a group with guanidinoacetic acid (GAA), and a group with zilpaterol hydrochloride (ZLH). We have revised and rewritten all relevant sections of the manuscript to make this point clearer and avoid any possible confusion.
- Comment: More than three lambs per treatment group is required for reliable statistical analyses.
Response: We recognize that a small sample size can limit the statistical robustness and reliability of the results. Considering this, we have decided to remove the corresponding section from the manuscript to maintain scientific rigor.
- Comment: Why was only the color of the meat measured and not other quality variables such as texture or drip loss?
Response: The aim of this study was to evaluate the effects of GAA and ZLH on growth performance, carcass characteristics, and basic meat quality indicators, with meat color selected as a key parameter due to its immediate impact on consumer acceptance and its feasibility within our available resources.
Results
- General response: We have reviewed and updated the tables in the manuscript to clearly define all variables included in the results section.
Discussion
- Comments: It should be creatine; creatinine is a byproduct of creatine metabolism and is primarily eliminated by the kidneys. Do you mean creatine?
Response: We would like to clarify that the biochemical analyzer used in this study measures serum creatinine concentration. The manuscript refers specifically to creatinine, as this was the analyte quantified in the samples.
Reviewer 2 Report
Comments and Suggestions for Authors
Estimated researchers, greetings,
Follow the suggestions below.
Line 22 - Summary: Add whether the animals were castrated or not castrated.
Add the statistical design.
I suggest adding the adaptation period of animals (14 days)
Material and Methods
I suggest adding some climate data from the experimental site if possible, such as temperature, air moisture.
I suggest adding the size of the bays that the animals were housed and type of floor.
Line 103 - Did the animals go through a fast before the initial weighing? How many days?
Was the fed supplied with calculation of how many percent leftovers? I suggest adding.
Add AOAC Methods (protocols) used, for example: Dry Matter (DM, Method 930.15), Ash (Method 942.05), Organic Matter (OM, Method 942.05), Crude Protein (CP, Method 954.01).
Line 172- If possible, the statistical model used (equation) must be added
Line 190 - Check if instead of 6 variables are not 5.
Table 3 - Check that the averaga daily gain of animals are correct
Super Add the sentence “In Meat Color Parameters, GAA and GAA + ZLH GROUPS DID NOT HAVE 230 Effects on Lightness, Redness, Yellowness, Chroma or Hue Angle of Lamb Meat (p> 0.05, table 5). Moreover, in Significant Differences Were Observed in the Contrasts (p> 0.05). “Between Table 4 and Table 5
262-268-Unlike the introduction, I suggest that in the discussion avoid work on pigs and birds, and SUD = bristitura for research with ruminants, goats, sheep, cattle. Since they have jobs using GAA and ZLH additives.
Conclusion: I suggest removing from the conclusion the doses used (0.2% DM) and (0.6 g/kg of DM) since in methodology and its results already describe well.
The sitting “However, an Increase In Serum Glu- 332 Cose and Creatinine Could Indicate that gaa additive influenced energy metabolism, that 333 could not be seen as beer productive paerns. Overall Results of the Present Work Could 334 Indicate that lengthen the Time of Study and/or Different Dosage of Additives Are Needed 335 in Order to Obtain The Expected Results. ” This is more for a discussion than for a conclusion.
SUGGESTION: I believe that the authors should not recommend additives because they have no effect on nutrient digestibility, carcass characteristics or changes in meat color and due to the effect on serum glucose and creatinine new studies are necessary.
Author Response
Summary
- Comments: Add whether the animals were castrated or not castrated. Add the statistical design. I suggest adding the adaptation period of animals (14 days)
Response: The Abstract has been revised to include the fact that the animals were non-castrated males, that a completely randomized design was used, and that animals underwent a 14-day adaptation period prior to the trial.
Material and methods
- Comments: I suggest adding some climate data from the experimental site if possible, such as temperature, air moisture. I suggest adding the size of the bays that the animals were housed and type of floor.
Response: We have included climatic information for the experimental site, including average annual temperature and rainfall, in the Materials and Methods section. Additionally, we specified that animals were housed in individual pens measuring 1.5 × 0.75 m with concrete floors.
- Comments: Line 103 - Did the animals go through a fast before the initial weighing? How many days? Was the fed supplied with calculation of how many percent leftovers? I suggest adding.
Response: We have clarified in the Materials and Methods section that animals were fasted for 24 hours before initial weighing. In addition, we specified that feed was supplied allowing approximately 10% refusal of dry matter intake to ensure ad libitum access.
- Comment: Add AOAC Methods (protocols) used, for example: Dry Matter (DM, Method 930.15), Ash (Method 942.05), Organic Matter (OM, Method 942.05), Crude Protein (CP, Method 954.01).
Response: We have revised the Chemical Analyses subsection to include the specific AOAC official methods used.
- Comment: Line 172- If possible, the statistical model used (equation) must be added
Response: The statistical model used for analyzing the data has been added to the Statistical Analysis section.
- Comment: Line 190 - Check if instead of 6 variables are not 5.
Response: The digestibility data were removed from the final version of the manuscript during revision.
Results
- Comment: Table 3 - Check that the average daily gain of animals is correct
Response: The ADG values in Table 2 are correct and correspond to least squares means adjusted by the covariate (initial body weight), not raw averages.
- Comment: Super Add the sentence “In Meat Color Parameters, GAA and GAA + ZLH GROUPS DID NOT HAVE 230 Effects on Lightness, Redness, Yellowness, Chroma or Hue Angle of Lamb Meat (p> 0.05, table 5). Moreover, in Significant Differences Were Observed in the Contrasts (p> 0.05). “Between Table 4 and Table 5.
Response: We have added the sentence summarizing the results of meat color parameters and contrasts immediately after Table 4, as requested.
- Comment: Unlike the introduction, I suggest that in the discussion avoid work on pigs and birds, and use studies based on research with ruminants, goats, sheep, cattle. Since there are studies using GAA and ZLH additives.
Response: We believe it is important to briefly mention studies in pigs and poultry to highlight the broader relevance and prior use of GAA and ZLH.
Conclusion
- Comments: I suggest removing from the conclusion the doses used (0.2% DM) and (0.6 g/kg of DM) since in methodology and its results already describe well. The sentence “However, an increase in serum glucose and creatinine could indicate that GAA additive influenced energy metabolism, that could not be seen as better productive patterns...” is more appropriate for the discussion than the conclusion.
Response: We have removed the specific doses from the Conclusion section to avoid redundancy. We agree with your comment and have moved the sentence from the Conclusion to the Discussion section.
Reviewer 3 Report
Comments and Suggestions for Authors
Manuscript titled “GAA and ZLH fed addition on lambs’ nutrient absorption, carcass, and blood chemistry” investigated the individual and combined effects of GAA and ZLH on lambs. The experimental design was novel and interesting. However, there are several issues that need addressing before considering it for publication.
General:
Title: I recommend revising the title to make it more clear and concise.
Abstract: The abstract needs a more detailed description of both the experimental design and results. It should provide key information that readers can easily understand. Additionally, there should be specific descriptions of the results related to nutrient absorption and carcass characteristics. It was also important for the current study.
Materials and Methods:
Why use different unites (0.2% DM GAA of diet, 6 mg/kg of DM ZLH for ZLH). Instead of grouping into control, GAA, and GAA+ZLH groups, consider including a separate ZLH group to better understand individual effects. Ensure the TMR (Total Mixed Ration) meets the nutritional requirements for lambs as recommended by NRC (National Research Council) or other relevant standards.
Was the feces collected for nutrient digestibility determination treated to account for biological nitrogen fixation?
Results: No difference in body weight dose it because of the unsimilar initial body weight of lambs?
Discussion: Discussion was too broad. It should be based on the current results.
Style of reference cited in text wasn’t correct, such as 31,32, 33 and 34.
Specific:
All p should be italic. P value was reserved to two decimal places in Tables. Ensure references cited in the text follow the correct format.
Line 24: Why use 0.2% GAA substituted TMD? GAA as an additive, it was usually used for additive in diets? Total mixed ration (TMR) not total mixed diet (TMD)
Line 95: LW?
Line 106-107: Additive addition method is unclear.
Line 109: How long did GAA used for lambs? Why ZLH used for the last 30 days of the experimental period?
Line 112: EE, Ca and P contents for diet?
Line 125-126: confused description
Line 168-171: Whole name of BUN. Serum samples were detected by semi-automatic machine. Why use kits to detect them?
Line 187-188: Confused
Table 3: Unites of FCR was blank. Unites of GLU is lack.
Comments on the Quality of English LanguageEnglish should be improved and submitted it into native English speaker.
Author Response
Title
- Comment: I recommend revising the title to make it clearer and more concise.
Response: We have revised the title to improve clarity and conciseness.
Abstract
- Comment: The abstract needs a more detailed description of both the experimental design and results. It should provide key information that readers can easily understand. Additionally, there should be specific descriptions of the results related to nutrient absorption and carcass characteristics. It was also important for the current study.
Response: We have revised the abstract to include a clearer description of the experimental design, as well as more specific results.
Material and methods
- Comment: Why use different unites (0.2% DM GAA of diet, 6 mg/kg of DM ZLH for ZLH). Instead of grouping into control, GAA, and GAA+ZLH groups, consider including a separate ZLH group to better understand individual effects. Ensure the TMR (Total Mixed Ration) meets the nutritional requirements for lambs as recommended by NRC (National Research Council) or other relevant standards.
Response: Different units were used for GAA (percentage of diet DM) and ZLH (mg/kg of DM) to reflect standard practices based on previous literature and manufacturer recommendations. Regarding treatment structure, the study was designed to evaluate the individual effects of GAA and ZLH, which is why a GAA+ZLH combination group was not included. As suggested, the TMR was formulated to meet the nutritional requirements for growing lambs based on NRC (2007) guidelines and is detailed in Table 1
- Comment: Was the feces collected for nutrient digestibility determination treated to account for biological nitrogen fixation?
Response: The nutrient digestibility section was initially considered but later removed during manuscript revision. Therefore, fecal collection and corrections for biological nitrogen fixation were not applicable.
Results
- Comment: No difference in body weight dose it because of the unsimilar initial body weight of lambs?
Response: Although there was some natural variation in initial body weight among lambs, this variable was included as a covariate in the statistical model to adjust for its potential influence. Therefore, the lack of differences in final body weight is not attributed to differences at baseline but rather to the limited effect of the additives at the tested dosages.
Discussion
- Comments: Discussion was too broad. It should be based on the current results. Style of reference cited in text wasn’t correct, such as 31,32, 33 and 34.
Response: The Discussion section has been revised to focus more directly on the findings of the present study. In addition, the in-text reference style has been corrected to follow journal formatting guidelines.
Specific:
- Comments: All p should be italic. P value was reserved to two decimal places in Tables. Ensure references cited in the text follow the correct format.
Response: We have revised the manuscript to ensure that all instances of p are italicized, p-values in tables are consistently reported with two decimal places, and in-text citations follow the correct format according to journal guidelines.
- Comment: Line 24: Why use 0.2% GAA substituted TMD? GAA as an additive, it was usually used for additive in diets? Total mixed ration (TMR) not total mixed diet (TMD).
Response: We have corrected the term “total mixed diet (TMD)” to “total mixed ration (TMR)” throughout the manuscript. The inclusion of GAA at 0.2% of diet DM follows previous studies in ruminants and reflects standard usage as a dietary additive rather than a substitute.
- Comment: Line 95: LW?
Response: The abbreviation “LW” has been removed, and the term is now written in full to improve clarity.
- Comment: Line 106-107: Additive addition method is unclear.
Response: We have clarified the method of additive inclusion in the Materials and Methods section.
- Comment: Line 109: How long did GAA used for lambs? Why ZLH used for the last 30 days of the experimental period?
Response: GAA was supplemented throughout the entire 60-day experimental period. In contrast, ZLH was included only during the last 30 days, following standard practice and manufacturer recommendations for β-adrenergic agonists, which are typically administered during the finishing phase to enhance muscle deposition.
- Comment: Line 112: EE, Ca and P contents for diet?
Response: Ether extract (EE), calcium (Ca), and phosphorus (P) values were not included in the original formulation table as they were not directly analyzed in this study. However, the total mixed ration was formulated using standard feed composition tables to meet the nutritional requirements for growing lambs, in accordance with NRC (2007) guidelines.
- Comment: Line 125-126: confused description
Response: The digestibility section was removed during manuscript revision, and any remaining reference to it in lines 125–126 has been clarified or deleted to avoid confusion.
- Comment: Line 168-171: Whole name of BUN. Serum samples were detected by semi-automatic machine. Why use kits to detect them?
Response: The full name of BUN (blood urea nitrogen) has been provided in the text. Serum metabolites were measured using commercial diagnostic kits in conjunction with a semi-automatic biochemical analyzer.
- Comment: Line 187-188: Confused
Response: The section related to nutrient digestibility was removed during the revision of the manuscript. Therefore, the content previously located at lines 187–188 is no longer present.
- Comment: Table 3: Units of FCR was blank. Units of GLU is lack.
Response: The units for feed conversion ratio (FCR) and glucose (GLU) have been added to for clarity and consistency. FCR is expressed as kg DMI/kg gain, and glucose concentration as mg/dL.
- Comment: Comments on the Quality of English Language: English should be improved and submitted to a native English speaker.
Response: The manuscript has been revised for grammar, clarity, and scientific language. Additional editing has been performed to improve fluency and readability.
Round 2
Reviewer 1 Report
Comments and Suggestions for Authors
The article has been improved, but some unclear points remain.

Author Response
Simple summary
- Comment: What does it mean "to en"?
Response: The phrase was a typographical mistake. We have corrected it to: "...to enhance lamb growth and meat production." (line 22)
Abstract
- Comment: This paragraph is not clear.
Response: We revised and adjusted all the abstract section for clarity. Also we adjusted the paragraph to a maximum of 200 words.
Introduction
- Comment: If GAA is known to increase shear force, why wasn't this measured in the study?
Response: Shear force was not included in this study due to logistical limitations and because the primary focus was on performance, carcass traits, and blood parameters. We acknowledge its relevance and will consider including shear force measurements in future research. Thank you for the comment.
Materials and methods
- Comment: Were blood samples collected from all the animals?
Response: Yes, blood samples were collected from all animals in each treatment group at five time points (days 1, 15, 30, 45, and 60). For clarity we modified the sentence in lines 218 and 219.
Results
- Comment: It should say "the treatments with additives instead of the GAA treatment and the control group."
Response: The comparison was between the additive treatments (GAA and ZLH) and the control group. We have revised the sentence accordingly to accurately reflect the experimental design and statistical contrasts (lines 264-267).
Discussion
- Comment: This paragraph is not clear (line 241-242)
Response: We agree that the sentence was ambiguous. The corrected version has been included in the revised manuscript (lines 336-340).
- Comment: The evaluation of VFA production was not conducted.
Response: We´re agree that the evaluation of VFA production would have provided valuable complementary information regarding the effects of GAA on ruminal fermentation. Although we referenced previous studies that reported increases in VFA concentrations with GAA supplementation, we did not directly measure VFA production in the present study due to logistical limitations. Thank you for your observation.
- Comment: Creatine or creatinine? Creatinine was measured in this study; however, GAA is not a precursor of creatinine, but rather of creatine.
Response: The GAA is a direct precursor of creatine, not creatinine. In this study, we measured serum creatinine as an indirect indicator of creatine metabolism. We have revised the manuscript to correct this terminology. Thank you for this important clarification (line 379).
Reviewer 2 Report
Comments and Suggestions for Authors
The authors have largely carried out a suggested review.
Thus, it is suitable for publication.
Author Response
we appreciate all your valuable observations and revisions
Reviewer 3 Report
Comments and Suggestions for Authors
Authors were well revised according to the comments.
L118: Ether Extract and NEg (Net energy for growth) should be provided.
L234-235 No bold words.
Author Response
Simple summary
- Comment: L234-235 No bold words.
Response: The bold words in lines 234–235 have been removed to comply with formatting standards.
- Comment: L118: Ether Extract and NEg (Net energy for growth) should be provided.
Response: Thank you for your valuable suggestion. We have now included the ether extract (EE) and metabolizable energy (ME) values. Instead of NEg, we opted to present the ME value.